# Recent Updates on the Significance of KRAS Mutations in Colorectal Cancer Biology

**DOI:** 10.3390/cells10030667

**Published:** 2021-03-17

**Authors:** Loretta László, Anita Kurilla, Tamás Takács, Gyöngyi Kudlik, Kitti Koprivanacz, László Buday, Virag Vas

**Affiliations:** 1Research Centre for Natural Sciences, Institute of Enzymology, 1051 Budapest, Hungary; laszlo.loretta@ttk.hu (L.L.); kurilla.anita@ttk.hu (A.K.); takacs.tamas@ttk.hu (T.T.); kudlik.gyongyi@ttk.hu (G.K.); koprivanacz.kitti@ttk.hu (K.K.); buday.laszlo@ttk.hu (L.B.); 2Department of Medical Chemistry, Semmelweis University Medical School, 1071 Budapest, Hungary

**Keywords:** KRAS, colorectal cancer, RAS signalling, cancer stem cells, RAS-driven metastasis, CRC with age

## Abstract

The most commonly mutated isoform of RAS among all cancer subtypes is KRAS. In this review, we focus on the special role of KRAS mutations in colorectal cancer (CRC), aiming to collect recent data on KRAS-driven enhanced cell signalling, in vitro and in vivo research models, and CRC development-related processes such as metastasis and cancer stem cell formation. We attempt to cover the diverse nature of the effects of KRAS mutations on age-related CRC development. As the incidence of CRC is rising in young adults, we have reviewed the driving forces of ageing-dependent CRC.

## 1. Introduction

In spite of early-stage detection screening methods and innovative tumour therapies for colorectal cancer (CRC), CRC is still a cancer type that is a leading cause of death worldwide. Several risk factors as well as epigenetic and genetic alterations are implicated in CRC development. In addition to APC, TP53, and NMR gene-inactivating mutations and BRAF and PIK3CA activating mutations, the acquisition of mutated forms of RAS oncogene represent an important trigger for CRC development.

The RAS family consists of membrane-associated small GTPases which play essential roles in cell survival, proliferation, and differentiation. There are four RAS protein isoforms in humans: HRAS, NRAS, and two splice variants, KRAS4A and KRAS4B [1]. The most commonly mutated RAS gene in CRC is KRAS [2].

RAS proteins comprise two regions, the G domain and the C terminal hypervariable region. G domain sequences are highly homologous, while the hypervariable regions (HVR) differ across the RAS isoforms [3]. The G domain is responsible for catalytic activity and effector binding, while the function of the HVR is mainly binding to the plasma membrane [4]. The HVR region, which is responsible for plasma membrane (PM) association, undergoes a series of posttranslational modifications (PTMs) required for it to bind the membrane. Membrane attachment is indispensable for cellular RAS function [5]. One of the PTMs required for membrane localisation is prenylation, which occurs on the CAAX sequence at the C terminal end of the protein (Figure 1). This modification results in farnesylation of the cysteine residue in the CAAX motif by farnesyltransferase (FTase), which promotes proteolytic cleavage of the last three amino acids (AAX) by RAS-converting enzyme 1 (RCE1) [6]. Next, a methyltransferase catalyses carboxyl methylation of the new C terminus at the cysteine residue remaining from the CAAX motif, and this reaction neutralises the negatively charged end of the polypeptide chain and increases its membrane affinity [6,7]. These reactions, together called CAAX processing, boost the membrane affinity of RAS; however, they alone are insufficient to support strong PM association. Palmitoylation of one or two cysteine residues in the HVR region can increase the membrane binding affinity [5] (Figure 1). One exception is the KRAS4B isoform, on which no palmitoylation occurs due to its polybasic region upstream of the CAAX motif. This special region consists of eight positively charged lysine residues that can form electrostatic bonds with the negatively charged phospholipid headgroups of the PM, and along with the prenylated C terminus, it is sufficient for membrane association [6,8]. Many attempts have been made to block posttranslational modification of the HVR, especially RAS prenylation, with varying levels of success. Currently, there are two class of drugs that are clinically approved and used for anticancer therapy via inhibition of protein prenylation. These drugs have proved to be ineffective against CRC [6]; however, there may still be hope since FGTI-2734, a recently developed peptide inhibitor, can disrupt KRAS membrane localisation in various human cancer cell lines, including the DLD1 colon cancer cell line [9].

In addition to prenylation and palmitoylation, many other PTMs can take place on RAS proteins (Figure 1). For example, phosphorylation of Ser181 of KRAS4B by PKC kinase can reduce the interaction of the protein with the PM via endocytic recycling [10]. The G domain has two other phosphorylation sites, Tyr32 and Tyr64, on which modifications can also downregulate RAS activity by inhibiting the binding affinity for its effectors [11]. There are other known posttranslational modifications on RAS proteins, e.g., polyubiquitination, which can target them for proteasomal degradation [12,13]. There are also many sites for mono- and polyubiquitination, but instead of downregulating RAS signalling, these modifications can enhance GTP binding and facilitate effector activation. Other PTMs, e.g., acetylation, sumoylation, ADP-ribosylation, glycosylation, and nitrosylation, are also assumed to influence KRAS regulation, but the exact functions are still poorly described [3,5] (Figure 1).

## 2. Mutant KRAS-Driven Enhanced Cell Signalling in CRC

Wild type KRAS functions as a controlled binary molecular switch, cycling between inactive and active signal-transducing conformations. Once KRAS is in its GTP-bound state, it undergoes structural changes that allow it to bind and cooperate with downstream signalling molecules. RAS signalling is prevented when GTP is hydrolysed and RAS is in its GDP-bound state. These two states are tightly regulated by GEFs (guanine nucleotide exchange factors) and GAPs (GTPase-activating proteins). GEFs fuel GDP release from RAS and facilitate GTP loading to activate RAS, while GAPs accelerate GTP hydrolysis, leading to RAS inactivation.

Essentially, the primary role of GTP-bound, active KRAS is to collect and activate multiple effector molecules at the membrane and to coordinate various signalling routes. In normal cells, upon ligand binding, receptor tyrosine kinases, e.g., EGFR, are autophosphorylated. Activated receptors then dimerise and recruit adaptor proteins (e.g., Grb2 or Shc) to their cytoplasmic tails. Via the adaptor proteins, GEFs are also associated with the molecular complex where they facilitate the conversion of inactive GDP-bound KRAS to the signal-transmitting, active GTP-bound form. Therefore, within KRAS molecules, the structure of the GTP/GDP interacting site determines its function.

The integrity of the GTP/GDP binding site in the KRAS G domain has special importance in maintaining the well-regulated function of KRAS, and a single amino acid change within this site can abolish normal regulation. Therefore, it is not surprising that CRC-associated mutations in KRAS are located within this effector site (Figure 2A). Single base substitutions in codons 12 and 13 are very common cancerogenic mutations that affect glycine residues in the GTP-binding pocket critical for GTPase function (Table 1). It is broadly accepted that these KRAS mutations lead to stabilisation of the protein in its prolonged active state, thereby amplifying the downstream signalling pathways.

The main and most robust KRAS-regulated signalling route is the MAPK pathway (Figure 2B). When KRAS is activated either traditionally via receptor activation or via oncogenic mutations, KRAS proteins dimerise. The KRAS dimers then bind and activate RAF kinases. Next, RAF can phosphorylate the two catalytic serine residues of MEK. MEK, a dual threonine and tyrosine recognition kinase, then phosphorylates other kinases in the pathway, namely ERK1 and ERK2. In normal cells, activated ERKs initiate multiple effector mechanisms, e.g., the G1/S-phase transition, inhibition of apoptosis, and cell motility [14]. Notably, in CRC cells, it has been demonstrated that KRAS mutations lead to aberrantly elevated MAPK signalling [15,16]. Therefore, many attempts have been made to target and inhibit molecules downstream of KRAS, although breakthrough success in therapeutic applications has not been realised [17].

Mutations in KRAS lead to increased proliferation of CRC cells, and in combination with other mutations, e.g., in the *APC* gene, promote tumorigenesis. Only in recent years has it been systematically described how concurrent misregulation of both MAPK and WNT signalling can lead to CRC. The key component of the crosstalk between the MAPK and WNT pathway is Glycogen synthase kinase 3β (GSK3β) [18]. GSK3β regulates proteasomal degradation of β-catenin and NRAS protein. GSK3β activation depends on APC and ERK activity; therefore, upon development of a loss-of-function mutation in APC, GSK3 no longer exerts its destabilising effect on β-catenin, and WNT signalling is enhanced [19]. When a KRAS mutation occurs along with an APC mutation during CRC development, mutant KRAS-driven MAPK signalling result in hyperphosphorylation of ERK, which further inhibits GSK3β function. Both malfunctioning APC and mutant KRAS-enhanced ERK activity synergistically inhibit GSK3-β to amplify the β-catenin effect [18]. Therefore, one possible therapeutic intervention against mutant KRAS would be to directly target GSK3β to increase its activation to deregulate β-catenin [20]. This type of trial has been recently performed by Lee et al., who treated CRC xenografts with small-molecule compounds to re-activate GSK3β-driven signalling [21].

**Figure 2 cells-10-00667-f002:**
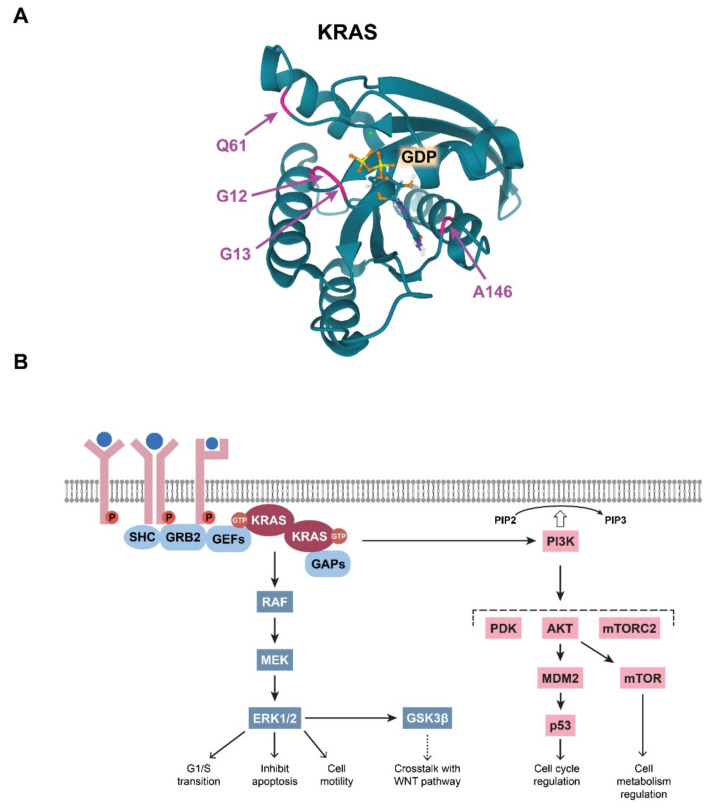
(**A**) Visualisation of the colorectal cancer (CRC)-related mutations in the KRAS G domain. The glutamine residue at position 61, the glycine residues at positions at 12 and 13, and the alanine residue at position 146, which form the GDP/GTP pocket, are indicated with colours. Image from the RCSB PDB (rcsb.org (accessed on 3 March 2021)) of PDB ID: 6MBT [22]. (**B**) Schematic representation of the summary of the RAS-driven signalling pathways.

**Table 1 cells-10-00667-t001:** An overview of KRAS mutations in CRC. Mutations detected frequently in CRC are in the upper part of the table, and the frequencies of the mutations are indicated based on http://www.cbioportal.org/ (accessed on 3 March 2021).

Amino Acid Position	Type of Amino Acid Substitution and Incidence in CRC	Alterations
G12	G12A (2.13%) [23]	survival rates are low with this mutation [24] the mutated protein signals via RAF via the ERK signalling pathway [24]
	G12C (3.05%) [23]	strong KRAS mutant [25], poor prognosis in CRC [26]
	G12D (13.17%) [23]	extensive hyperplasia, typified by lengthening of the crypts in in vivo mouse models [25], weak driver [27] mutated protein interacts with the PI3K/AKT, JNK, p38, and FAK pathways [24]
	G12V (9.13%) [23]	poor prognosis in CRC [28], aggressive cancer phenotype [24], the mutated protein acts via the ERK pathway and participates in RAF signalling [24]
G13	G13D (7.31%) [23]	sensitive to EGFR inhibition, medium hyperplasia in in vivo mouse models [25], increased glucose uptake and lactate production [29]
A146	A146T (2.42%) [23]	often appears with other mutations, e.g., MAPK pathway mutations [25]
G12	G12S	enhanced RAF activation [30]
	G12R	increased glutaminolysis [31]
G13	G13A	insensitive to MEK inhibitor [32]
Q61	Q61L	GTP hydrolysis is reduced [33], transition state is unstable [33], RAF affinity is increased [33], interacts with the RAF kinase pathway [33], co-occurs with BRAF mutations [33]
	Q61H	attenuated HVR-G domain association of KRAS [30]
A146	A146P	in vitro EGFR or MEK inhibitors are effective [34]
	A146V	this mutation is a sign of resistance to EGFR inhibitors [34] might have a predictive role for relapse-free survival [33]

In addition to the positive feedback amplification of MAPK signalling and WNT/β-catenin activation, mutant KRAS also exerts its proliferation-stimulating effect via the PI3K/AKT pathway (Figure 2B). The mutational status of PI3K is a potential prognostic marker in CRC as mutations in the catalytic alpha subunit of PI3K are frequently associated with poor prognosis-associated KRAS mutations in CRC [35]. Activated wild type KRAS and constitutively activated mutant KRAS directly bind PI3K. As PI3K is activated by RAS, it generates PIP_3_ secondary lipid messengers in the membrane. This signal can be propagated through several downstream effector molecules via PIP_3_. The KRAS-driven PI3K-generated increase in the PIP_3_ concentration in the membrane leads to the recruitment of PDK, AKT, and mTORC2 as well as activation of its target proteins. AKT (alternatively, protein kinase B (PKB)) is a serine/threonine specific kinase, and in its activated form, it can interact with more than 100 binding partners.

Recently, two AKT partner molecules, specifically MDM2 and the mechanistic target of rapamycin (mTOR) complex, have been highlighted as important elements in KRAS-driven CRC (Figure 2B). It is known that MDM2 directly interacts with p53, thereby inhibiting p53′s cell cycle regulatory function and its tumour suppressor activity as well as promoting its degradation. Therefore, MDM2 inhibition may hold clinical promise, particularly in combination with MEK inhibition, as Hata et al. recently demonstrated in a KRAS-mutant CRC model [36]. Among the AKT binding partners, the mTOR complex has emerged as a promising therapeutic target. Pathways initialised by mTOR regulate normal metabolism, and in CRC cells, glycolysis is aberrantly upregulated and glucose consumption is elevated.

Therefore, it is tempting to speculate that altering mTOR function may limit CRC viability. Consistent with the idea, Fritsche-Guenther at el. compared the effects of targeting mTOR in KRAS- and BRAF-mutant CRC lines and concluded that oncogene-specific mutations account for alterations in the resistance mechanisms and changes in CRC metabolic phenotypes [37].

The interdependence of RAS-mediated signal routes represents only one aspect of the complexity of KRAS signalling, and the discovery of microRNA-driven epigenetic regulation of KRAS expression further increased the complexity of known KRAS biology. The heterogeneous group of microRNAs (miRNAs) is composed of small, single-stranded, non-coding RNA molecules. Currently, several miRNAs (let-7, miR-193, miR-143, miR-18a) that target the KRAS UTR-region, leading to KRAS mRNA degradation and/or translational repression, have been identified [38,39,40,41]. Consistent with their function, it was found that the levels of KRAS-targeting miRNAs are decreased in CRC, highlighting miRNAs as potential diagnostic or prognostic biomarkers either as single factors or in panels of miRNAs.

The study of KRAS-driven cell signalling and the elucidation of an overarching concept of the entire process are technically daunting undertakings. Therefore, several experimental model systems have been developed during recent decades with the aim of studying specific protein–protein interactions or other aspects of KRAS signalling. To understand the mechanisms of the signalling pathways initiated by KRAS mutations, approaches based on in vitro cell lines and in vivo mouse models have been used (Table 2 and Table 3). The majority of our knowledge about CRC pathogenesis and mutant KRAS signalling has been generated using various mouse models that recapitulate phenotypes of KRAS-mutant CRC patients (Table 3). By using KRAS-mutant human CRC cell lines, we have been able to understand the effects of several specific KRAS mutations and to rapidly test newly developed drug candidates.

## 3. Role of KRAS in CRC Stem Cells

Growing evidence suggests that a subset of cells within tumours, i.e., cancer stem cells (CSCs), possess stem cell properties such as self-renewal and differentiation drive heterogeneity within the tumour while also promoting cancer growth, proliferation, resistance, invasion, and metastasis [57]. This special cell type is, therefore, a promising target in the development of cancer therapies. To date, there are several hypotheses for how CRC develops [58]. According to the “top-down” hypothesis, more differentiated luminal cells dedifferentiate and regain stem cell properties, thereafter functioning as CSCs. The “bottom-up” hypothesis, however, argues that CRC originates from crypt base intestinal stem cells (ISCs). In both models, due to deregulated WNT signalling, initiating cells escape regulation and become CSCs [59]. Genetic alterations in both the WNT/β-catenin [21,60,61] and EGF/RAS [21,62,63,64] pathways drive CRC progression by influencing ISC self-renewal and proliferation, respectively, resulting in an abnormally sustained stem cell state. Initiating mutations, e.g., loss-of-function mutations in APC, cause robust WNT signalling deregulation [61], and when accompanied by mutations in other genes, e.g., KRAS (or components of TGFB, PI3 kinase, and TP53 signalling), can lead to malignant transformation [65,66].

KRAS is an important modulator of CSCs since its downstream effectors, the MAPK/ERK and PI3K/AKT pathways, are regulators of WNT/ β-catenin [65,67] signalling. However, an activating mutation in KRAS alone is insufficient to initiate CSCs [65,68]. As discussed above, APC mutations paired with oncogenic KRAS are activators of CSCs, driving malignant transformation and metastasis. The underlying mechanism was shown to be that APC loss leads to both nuclear accumulation of β-catenin and stabilisation of oncogenic KRAS which, then, via the ERK pathway, further enhances and fuels activated WNT/β-catenin signalling and establishment of CSC characteristics, including enhanced sphere-forming capacity, tumour size and weight, and expression of CSC markers (e.g., CD44, CD133, and CD166) [65,69]. The importance of KRAS stability in CSC activation was also demonstrated by work on WD repeat protein 76 (WDR76), an E3 ligase that destabilises RAS, thereby acting as a tumour suppressor [70]. Ro et al. demonstrated that in APC-mutant CRC tumours, cytosolic WDR76 acts as a tumour suppressor by binding to and degrading KRAS via polyubiquitination-dependent proteasomal degradation, ultimately causing decreased activation of the WNT/β-catenin pathway. WDR76 deficiency increased oncogenic KRAS, ERK, AKT, and β-catenin levels and enhanced the expression of the CSC markers LGR5, CD44, CD133, and CD166 in APC-mutant DLD-1 CRC cells [70]. Hwang et al. identified REG4 as the most significant mediator (among other genes such as *IL8*, *PHLDA1*, *S100A4*, *S100A6*, and *PROM1*) in the promotion of CSC properties by oncogenic KRAS. REG4 was found to act through WNT/β-catenin signalling at the level of receptor activation in KRAS/APC double-mutant CRC cells [64].

In colon cells, LGR5, a WNT signalling component, initiates an intestinal stem cell (ISC) program; however, abnormal LGR5 expression does not initiate premalignant colon adenomas unless other genes, e.g., *APC* or *CTNNB1* (β-catenin), are also mutated to drive aberrant WNT signalling. The progression from a premalignant state to malignant status can be fuelled by oncogenic KRAS [71]. Le Rolle et al. recently demonstrated an alternative de-differentiation model for colon adenoma progression into CRC. They compared gene expression in stage I colon carcinomas with that in benign colon adenomas in human colon tissue and showed that while LGR5 expression correlated with an ISC signature in wild type KRAS (stage I) colon carcinomas, the presence of oncogenic KRAS, regardless of LGR5 expression, was associated with an embryonic stem cell (ESC)-like transcriptional signature instead of an ISC program. This ability of oncogenic KRAS was demonstrated in mutant KRAS-transfected SW48 human colon cancer cells as well, where the induction of an ESC-like program was indicated by upregulation of factors required for reprogramming and ESC and colon cancer maintenance, e.g., SOX2, FGFR1, LCK, validated at both the mRNA and protein levels. Increased soft agar colony growth and correspondingly increased in vivo tumour growth were also observed. Such an ESC-like program is foreign to colon cells, representing a distinct route for CRC formation in the presence of oncogenic KRAS. miR145, an inhibitor of the embryonic stem cell program, was found to counteract the effects of mutant KRAS by suppressing malignant growth and promoting the differentiation of mutant KRAS colon cancer cells [72].

As discussed above, oncogenic KRAS is a key driver of CRC progression from premalignant colon adenomas to stage I colon carcinomas, and it acts as an activator of CSCs in tumours. The expression levels of the CSC markers CD44 and CD166 correlate with higher risk of lymph node involvement and liver and lung metastasis in CRC patients carrying KRAS mutations [73]. Thus, it is of great importance to develop new strategies for targeting oncogenic KRAS and its related pathways in the context of CSC activation and maintenance.

## 4. The Relevance of CRC Laterality in Light of KRAS and BRAF Mutations

CRC is a molecularly and pathologically heterogeneous disease. Various subtypes can be distinguished based on the genetic alterations, e.g., the KRAS or BRAF mutational status, or based on the location of the primary CRC, i.e., originating from the right or left side. These features represent potential prognostic and predictive cancer biomarkers [74].

Interestingly, only a few cases have been reported where KRAS and BRAF mutations were simultaneously present, and in general, mutations in the two genes are mutually exclusive in CRC. The reason for this phenomenon is that KRAS and BRAF are members of the same signalling pathway, i.e., the MAPK pathway (Figure 2B). Mutations in BRAF lead to RAS-independent BRAF activation or increased kinase activity. As the first steps of the MAPK pathway involve GTP-KRAS dimerisation and BRAF activation, either a KRAS or BRAF oncogenic mutation is sufficient for tumourigenic effects and enhanced MAPK signalling (Figure 2B) [75]. BRAF mutations lead to RAS-independent BRAF activation or increased kinase activity.

A shared property of KRAS- and BRAF-mutant CRCs is that both types are generally unaffected by EGFR inhibitor therapy since both mutations transmit the EGFR signal independently of receptor activation. Furthermore, primary CRC with KRAS or BRAF mutations are more likely to occur on the right side [74]. The observation that right-sided CRC has a worse prognosis compared to the left-sided tumours has been described earlier, and KRAS and BRAF mutations have recently also been established as negative prognostic cancer markers [76,77]. To translate these findings into clinically relevant information, more comprehensive data analysis is needed to use these markers not only in defining the prognosis of the patients but also in the selection of the most effective therapeutic regime for CRC patients.

## 5. Metastasis during CRC Development

Although CRC does not represent a single pathological entity, the prevalence of metastasis among the heterogenous groups of CRC patients is very high. Several studies have demonstrated that the most common metastatic sites of CRC are the liver and lungs, while less commonly, metastases develop at the peritoneum, distant lymph nodes, or in bone [78,79]. Furthermore, lung metastases frequently develop together with liver metastases [80]. Interestingly, the origin of CRCs differentially predestines the site of metastasis in patients. The combined data of two large patient cohorts showed that lung metastasis is more frequent in rectal than in colon-originated cancers (Figure 3a) [80,81]. Lung metastases have better overall survival and slower growth than liver metastases [82]. Consequently, it was demonstrated that rectal cancer patients have better overall survival compared with colon cancer patients [83,84].

The invasion–metastasis cascade of cancer cells involves several steps, beginning with the penetration of tumour cells into surrounding matrix followed by transfer into the circulatory system, such as blood or lymphatic vessels. The metastasis cascade continues with the transportation of tumour cells followed by extravasation into distant tissues. The process ends with the colonisation of distant organs and the development of new tumours [87]. In CRC cells, epithelial–mesenchymal transition (EMT) is the first step in the cascade, which includes several biological processes, e.g., enhanced motility and increased degradation of the extracellular matrix (ECM) as well as elevated resistance to cell death [88]. During liver metastasis, platelets and neutrophils can chemically protect circulating tumour cells (CTCs) from various stresses. Jiang et al. demonstrated that a physical interaction between platelets and CTCs protects tumour cells against killer cell-mediated lysis and provides the CTCs in the circulatory system with a “pseudonormal” phenotype [89]. By contrast, the tumour-promoting role of neutrophils is not so clear in each situation. Several studies confirmed neutrophil activity as a tumour progression marker [90], while others observed antitumour activity [91]. Collectively, neutrophils can act as a prognostic factor depending on tumour type and disease stage [92]. Previous work demonstrated that angiogenesis has an important role in liver-specific metastasis [93]. Angiogenesis can support the growth of tumour cells and can also help them to enter the circulatory system via the formation new blood vessels [94]. Accumulating data have confirmed that KRAS activating mutations have an important role in angiogenesis [95] as they activate vascular endothelial growth factor (VEGF), a key mediator of angiogenesis.

The metastatic path of colorectal tumour cells to distant organs in the body is generally unclear [96], however, recent progress in tumour biology has shed light on which cell types within CRC contribute to metastasis formation. The level of circulating tumour cells (CTCs) is an accepted prognostic marker as their presence in the bloodstream is associated with a lower survival rate [97]. Previous studies have reported RNA profiles of the most relevant CTC-specific genes associated with cell mobility, EMT, apoptosis, as well as cell signalling and interaction, further supporting the role of CTCs in metastasis [98,99]. The neutrophil–CTC connection was also confirmed by Szczerba et al., 2019, as CTC–neutrophil-associated gene clusters are highly efficient metastatic predictors.

Another prognostic marker of metastatic events is the level of circulating tumour DNA (ctDNA). ctDNAs are approximately 200 base-pair-long DNA fragments present as a small part of the total cell-free DNA fraction in the plasma. The level of ctDNA in blood is correlated with the stage of CRC and tumour size [100]. ctDNA is released from apoptotic, active, or circulating tumour cells in the blood system. Analysis of the KRAS mutational status in the ctDNA population might also represent a useful tool for the identification of the mutations present among the CRC cells. With this method, it is easy to detect the level of mutant *RAS* ctDNA before and after therapy and to monitor the effectiveness of the treatment correlating with the disappearance of mutant *RAS* ctDNA [101]. As the major limitation of anti-EGFR therapy is the emergence of resistant clones, monitoring of mutant *RAS* ctDNA can provide information about the clonal heterogeneity of the CRC population. Van Emburgh et al. analysed the clonal evolution of CRCs during EGFR inhibitor treatment using ctDNA analysis and concluded that before therapy, several RAS mutant subclones co-existed in CRC cell populations, and that during EGFR blockade, the clonal composition of CRCs changed [102]. Some RAS-mutant clones disappeared, and a small subset of drug-resistant clones were selected under the stress of therapy, leading to disease recurrence. In general, genomic instability in CRC cells ensures the presence of latent clonal heterogeneity in the tumour. When CRC patients are treated with anti-EGFR therapy, the drug-sensitive clones are eliminated and mutant ctDNA becomes undetectable. However, under selective pressure imposed by the treatment, some pre-existing or newly acquired RAS-mutant CRC clones are positively selected. The detection of mutant *RAS* ctDNA following treatment could indicate the initial phase of disease and the emergence of drug-resistant clones, which would indicate the timeliness of urgent therapeutic intervention [103]. The ctDNA can be extracted quickly from blood plasma and represents a non-invasive and efficient method as it was confirmed that ctDNA is an ideal marker for diagnosis, detection of early disease recurrence, treatment response, and therapeutic resistance monitoring in CRC [104,105,106]. Using this ctDNA-based molecular tool, the length and intervals between anti-EGFR therapies in CRC could potentially be personalised.

Other studies have also reported interesting connections between mutational status and metastatic CRC (mCRC). For example, Pereira et al. showed that among mCRC patients, the KRAS gene mutation status is an important determinant of the site of metastasis formation [107]. More specifically, mutant KRAS carriers had a higher rate of lung metastasis (70%) than did patients with wild type KRAS (59%) [108]. Other studies have also suggested that patients whose primary tumour carried a KRAS mutation are more likely to develop lung metastasis during disease progression [109,110].

Various KRAS mutations have distinct consequences for CRC metastasis formation as specific single nucleotide substitutions in KRAS tend to activate colonisation of various metastatic sites in CRC patients (Figure 3b). Previous studies analysed the relationship between the location of metastases and specific mutation sites in KRAS [85,86]. According to Jones et al., codon changes in the 12th and 13th positions lead to more active KRAS protein, and metastatic CRC patients with these mutations have worse overall survival [86]. As shown in Figure 3b, KRAS mutations led to an overall higher rate of metastasis compared with CRC patients with wild type KRAS. CRC patients carrying a codon-13 mutation more frequently develop liver-only metastases compared with those carrying codon-12 mutations. Furthermore, multiple metastases (liver and lung) seemed to occur more often in a cohort carrying the codon-12 KRAS mutation (32.98%) than in patients with a codon-13 mutation or with no mutation (19.27% and 20.96%, respectively).

## 6. Association of Age with Disease Progression in Colorectal Cancer

In addition to the well-described role of RAS effector molecules in cell proliferation and survival, ageing is also connected to RAS-regulated downstream pathways. In particular, the abovementioned RAS-driven PI3K/AKT and ERK pathways (Figure 2B), which have a synergistic effect on the mTOR complex in cells, have longevity-regulating roles. A key regulatory function of the mTOR complex is the modulation of nutrient signalling, which affects longevity. Indeed, age is also an important factor in cancer formation, and recently, age-associated features of RAS-driven CRC have also been studied [111]. Based on the age at diagnosis, CRC can be divided into early- and late-onset disease (diagnosed before age 50 or after age 50, respectively) [112].

It is well known that CRC is more common in elderly patients, although retrospective analyses have shown that the incidence of CRC among young patients has tended to increase in recent decades [113,114,115,116,117]. Over fifteen years, there has been a nearly 90% increase in incidence estimated among patients aged 18–39 years [118]. The increasing trend in CRC frequency in young adults has not been fully explained. According to recent studies, several modifiable (environmental and behavioural) or non-modifiable (age, sex, genetic) factors appear to be associated with increased CRC risk in the young population (Figure 4).

With respect to early-onset CRC, modifiable risk factors such as obesity, Western diet (a high intake of red meat and low intake of fruit and vegetables), stress, antibiotics, and physical inactivity, together with sedentary behaviour, can all contribute to the increased CRC incidence in young adults [119,120,121,122]. Moreover, gut microbiome dysbiosis also has a strong effect on disease development, and alterations in the intestinal microbiota and decreased production of short chain fatty acids (SCFAs) appear to be relevant risk factors (Figure 4). Gomes et al. suggested that increasing the level of SCFAs by manipulating the colon microbiota has the potential to prevent or potentially treat CRC [123]. Another population-based cohort study (n = 68,860) identified non-modifiable risk factors in early-onset CRC development, such as family or personal history of polyps and rectal bleeding at the time of diagnosis [124]. In addition, genetic factors, e.g., hereditary CRC syndromes or CRC in first-degree relatives, are associated with higher CRC risk. Inflammatory bowel disease and additional modifiable risks, such as increased body mass index and smoking, have been shown to be moderate risk factors [125]. It has been hypothesised that the increasing trend of early-onset CRC is due to a yet undetected hereditary CRC syndrome, which might be present in the population [126]. An additional clinically relevant and the most easily modifiable risk is that young people without major symptoms do not participate in disease screening. In addition, annual screenings are not available for free for young adult as it they are for the elderly [127]. Early diagnosis of CRC in elderly patients may contribute to the decreasing incidence and decreasing mortality rate of late-onset CRC [128]. At the same time, several authors have proposed that the abovementioned risk factors underlay the increased incidence of early-onset CRC and late diagnoses, further aggravating progression to advanced disease stages. Therefore, authors have suggested that late diagnoses are associated with increased tumour development and more advanced stages among young patients [129] (Figure 4). In line with disease progression, a study revealed that younger patients have more aggressive tumour characteristics, although they also have a better survival rate as they can tolerate treatments [130].

The frequency of KRAS mutations in young and aged cohorts has been previously determined by several groups; however, the results are not conclusive, and the discrepancies among studies depend mainly on populational differences. Watson et al. presented a higher frequency of KRAS mutations in early-onset CRC, although according to Escobar et al., the frequency of KRAS mutations is similar in early-onset and in age-related CRC [131,132]. As mutant KRAS patients more frequently develop metastasis, it can be hypothesised that in specific age groups where KRAS mutations are overrepresented, a higher rate of metastasis can occur in these cohorts. However, based on several reports, the correlation of age, metastatic events, and the presence of KRAS mutations are contradictory. Among metastatic CRC patients, KRAS mutations are more often associated with old age [133]. It was also shown that among non-metastatic CRC patients, KRAS mutations are associated neither with age nor with liver–lung metastasis [134]. To further analyse the associations between age, mutant KRAS, and metastatic rate, our group used data from TCGA (The Cancer Genome Atlas) database. Colon adenocarcinoma and rectum adenocarcinoma patients were selected and classified by age (<66 or ≥66 years) and by metastasis rate (M0 = non-metastatic; M1 = metastatic) (Figure 5). Interestingly, the metastasis rate in a cohort of mutant KRAS patients older than 66 years was twice the rate of patients with wild type KRAS (16.42% vs. 8.82%) (Figure 5). Consistent with other age group-related observations, younger patients more frequently develop metastasis independent of their KRAS phenotype (Figure 5). Furthermore, younger patients carrying KRAS mutations more frequently develop metastasis than wild type KRAS patients from the same age group. As mentioned earlier, lack of screening might result in advanced-stage diagnosis in younger patients. Therefore, untreated disease could lead to poorer prognoses, and the presence of KRAS mutations could be an aggravating factor as it could affect metastatic events via altered cell signalling pathways.

## Figures and Tables

**Figure 1 cells-10-00667-f001:**
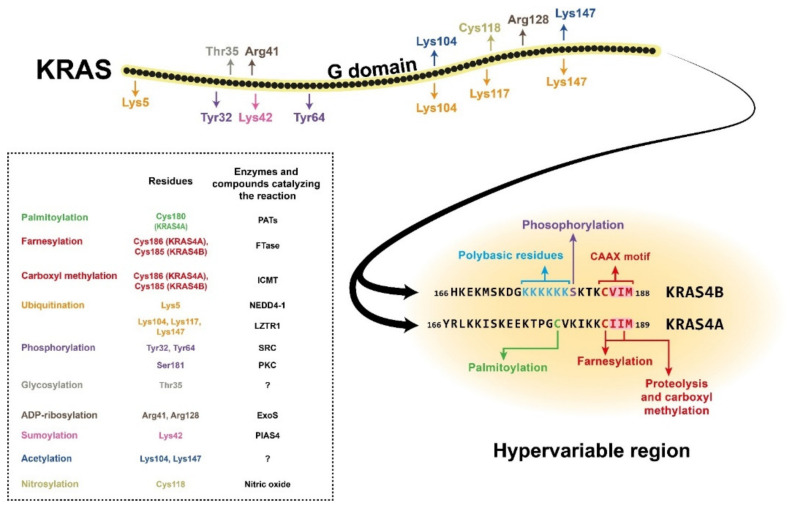
Structure of the KRAS4A and KRAS4B isoforms, showing the amino acid positions of posttranslational modifications with different colours. The hypervariable region sequences are presented separately for the two isoforms due to their high variability. The CAAX motifs and the polybasic residues on KRAS4B are also colour-coded. The posttranslational modifications are listed in the box with the affected amino acid sites and (where known) the enzyme or compound responsible for the reaction. NEDD4-1, neural precursor cell expressed developmentally downregulated 4-1; LZTR1, leucine-zipper-like transcriptional regulator 1; SRC, proto-oncogene tyrosine-protein kinase Src; PKC, protein kinase C; ExoS, exoenzyme S; PIAS4, protein inhibitor of activated STAT protein 4; PATs, protein acetyltransferases; FTase, farnesyltransferase; ICMT, isoprenylcysteine carboxyl methyltransferase.

**Figure 3 cells-10-00667-f003:**
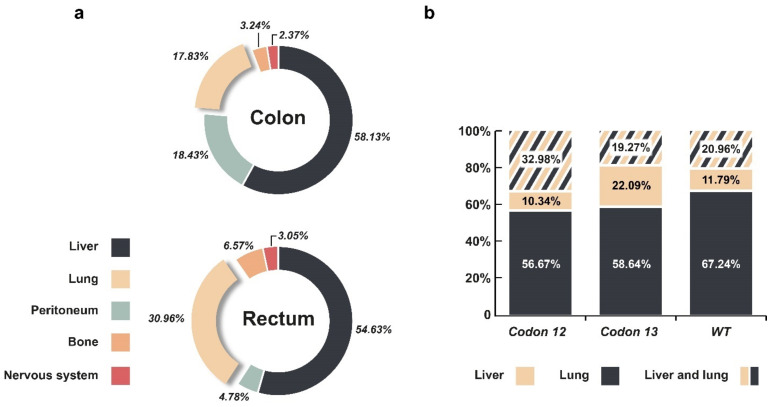
(**a**) Frequency of metastasis sites in CRC patients. Distributions of the metastatic sites in colorectal cancer patients (n = 27,506) were extracted from cohort data of Riihimaki et al. and Holch et al. [80,81]. (**b**) Frequency of single (liver or lung) metastasis and the combined appearance (liver and lung) metastasis among codon-12 and codon-13 mutant and wild type KRAS patients. Raw cohort data from He et al., 2020 and Jones et al., 2017 were normalised and pooled [85,86].

**Figure 4 cells-10-00667-f004:**
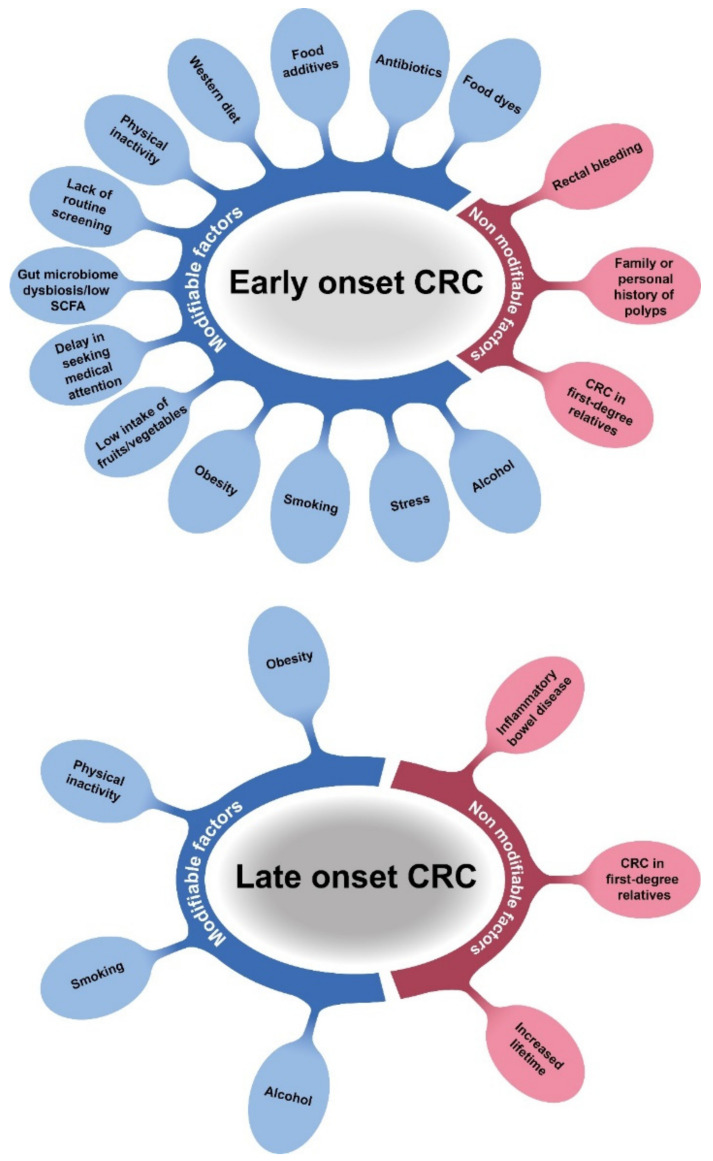
Modifiable and non-modifiable risk factors associated with early- or late-onset CRC.

**Figure 5 cells-10-00667-f005:**
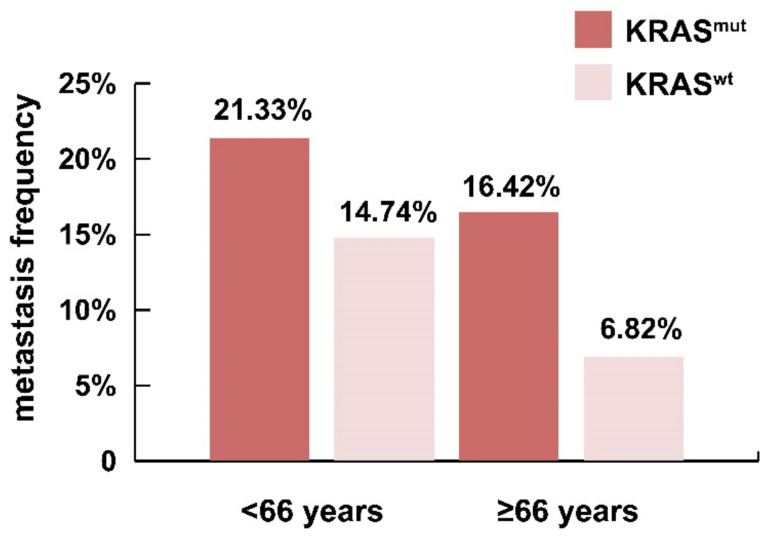
Prevalence of metastatic status among wild type and mutant KRAS CRC patients (n = 142–183). Metastasis is more frequent in mutant KRAS patients in both (<66 and ≥66 years) age cohorts. CRC patients carrying KRAS mutations (183 KRAS^wt^ and 142 KRAS^mut^ patients) and metastatic stage (M0 or M1) were extracted from PANCAN TCGA COADREAD (colorectal cancer) database on the Xena platform [135].

**Table 2 cells-10-00667-t002:** Summary of the in vitro experimental model systems currently used in CRC research showing that CRC cell lines harbouring specific KRAS mutations are useful tools for drug screening and for modelling the development of therapeutic resistance.

Cell Line	KRAS Mutation Position
HCT116	G13D [42,43]
HKe3	G13D [42]
HT29	G12D mutation created by lentivirus vector [44]
HCT15	G13D [43]
DLD-1	G13D [45]
LoVo	G13D, V14A [45]
SW620	G12V [43]
CCL187	G12D [46]
LS174T	G12D [46]
SW480	G12V [45]
CCCL23	A146T [47]
HCC2998	A146T [47]
LS1034	A146T [47]
CCCL-18	A146T [47]
HCT8	G13D [48]
CaCo2	doxycycline-inducible G12V mutation [37]

**Table 3 cells-10-00667-t003:** Summary of the in vivo experimental model systems currently used in CRC research for understanding mutant KRAS-driven signalling in mice.

Animal Models	Description of the Model	Major Conclusion
iKAP mice	Dox-inducible oncogenic Tet-Kras-G12D allele (Krasmut), null alleles of Apc and Trp53 (iKAP) [49]	metastatic CRC model: suitable model for the major genetic modifications that occur in CRC and for confirmation that KRAS mutation promotes tumour invasion and metastatic processes
APC-KRAS G12D mice	Tamoxifen-induced Cre recombinase, which cause APC loss and activation of Kras G12D mutation [50]	provides information on how the IL-8 cytokine affects KRAS-mutant CRC
Swiss female nu/nu mice	SW48 cells (expressing KRAS G13D or G12V produced by adeno-associated virus) injected into cecum [51]	metastatic CRC model: the KRAS G12V mutation is more aggressive than the KRASG13D mutation: more metastatic events, increased tumour cell survival, enhanced invasion
Nu/Nu female mice	KRAS G13D mutant CRC model: patient derived xenograft (PDX) injected into the right flank [52]	examines the effect of cetuximab cancer therapy: influence on tumour suppression, possible contribution to resistance
BALB/c-nu/nu nude mice	first model: intra-splenic injection of luciferase expressing HCT116 cellssecond model: tumour tissue insertion into cecum [53]	compares two xenograft mouse models in CRC and provides information on which mouse model researchers should use depending on their aim
C57BL/6 N mice	MC38-MR and CT26-MR (MEK inhibitor resistance) cells subcutaneously injected into the right flank of C57BL/6 mice [54]	examines the effect of combined treatment (MEK inhibitor, EGFR inhibitor, and PD-L1 inhibitor) on tumours harbouring KRAS mutations
BALB/c mice	MC38-MR and CT26-MR cells subcutaneously injected to the right flank of BALB/c mice [54]	examines the effect of combined treatment (MEK inhibitor, EGFR inhibitor, and PD-L1 inhibitor) on tumours harbouring KRAS mutations
BALB/c nude mice	implanted CRC tumour fragments into the subcutaneous layer: PDX (KRASG12D, G12V) [55]	defines possible new anti-EGFR treatment for KRAS-mutant CRC
BALB/c nude mice	SW480 cell suspensions injected subcutaneously into the left flank [56]	shows that 3-bromopyruvate can prevent tumour growth and cause cell death in a KRAS-mutant xenograft model
BALB/c male mice	azoxymethane-induced colon cancer [39]	shows how certain probiotics affect the expression levels of miRNAs and their target genes (KRAS, PTEN)

## Data Availability

The frequencies of the mutations are indicated based on http://www.cbioportal.org/ (accessed on 3 March 2021), and the TCGA (The Cancer Genome Atlas) was used to generate Figure 5 and data was extracted from PANCAN TCGA COADREAD (colorectal cancer) database on the Xena platform [135].

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
