# Peer review of "Recent Updates on the Significance of KRAS Mutations in Colorectal Cancer Biology"

_cells, 2021, doi:10.3390/cells10030667_

Round 1

Reviewer 1 Report

Dear author , 

I have appreciated your work even if I suggest to evaluate/improve these aspects:

  1. How target therapy affects the presence of mutating RAS clones during treatment
  2. I would replace figure 2 with a figure that summarize the way of RAS and as its constitutive activation has effect on other cellular pathways (indicating perhaps possible other molecular targets that could override the mutation RAS).
  3. Add a paragraph on the role of BRAF and its interaction with the RAS route in relation to the sidedness ( recent article on molecular difference between colon cancer, especially BRAF in right-sided colon, or in transverse colon )

Author Response

Answers are attached.

Reviewer 2 Report

This is a well writing manuscript. Authors describe the role of KRAS mutations in colorectal cancer. They also collect recent data on KRAS-driven cell signalling, in vitro/vivo models, and KRAS mediated CRC metastasis regulation. and cancer stem cell formation. Even though the general idea is interesting, it will be better if the authors prove the following:
1.    It will be better if authors discuss the difference in these KRAS-driven cell signalling more
2.    The association between Kras, their mutation and age in colorectal cancer is not clear describe in line 357-377
3.    Modifiable and non-modifiable risk factors associated with early- or late-onset CRC. Risks are not well discuss

Author Response

Answers are attached.

Reviewer 3 Report

In this manuscript, László and colleagues aimed to summarize recent data on the significance of KRAS mutations in colorectal cancer biology.

This review is quite interesting and provided an extensive number of published studies.

The manuscript is well written and comprehensive and English language and style are fine.

The only major modifications I would like to suggest are:

  1. Please provide for the session number 2 a scheme or a figure that represent the cell signaling pathways (MAPK and PI3K/AKT) involved in CRC.
  2. In Table 2 concerning the in vitro and in vivo experimental model systems used in CRC please add a new column that presented the major conclusions of the articles reported.
  3. In the session number 4 could be also interesting to introduce the concept of ctDNA (circulating tumor DNA) in parallel to the CTCs.
  4. In the text at page 10 line 372 it is mentioned the Figure 5 but it is missing. There is also at page 11 the figure legend of the “Figure 5”.

Minor concerns:

-Concerning the Figure 3b at page 9 (lines 312-314) it  is reported “ CRC patients carrying a codon 13 mutation more frequently develop lung metastasis compared with those carrying a codon 12 mutation”.  However, the percentage showed in the figure are almost similar. In my opinion, the major difference from the 2 cohorts is that “CRC patients carrying a codon 13 mutation more frequently develop LIVER metastasis compared with those carrying a codon 12 mutation”

-At page 11 in the figure legend it is mentioned “risk Figure 107”. It is probably a typo.

Author Response

Answers are attached.

Round 2

Reviewer 3 Report

I appreciated the effort of the authors to answer to my requests